# Proteome and Dihydrorhodamine Profiling of Bronchoalveolar Lavage in Patients with Chronic Pulmonary Aspergillosis

**DOI:** 10.3390/jof10050314

**Published:** 2024-04-25

**Authors:** Kristian Assing, Christian B. Laursen, Amanda Jessica Campbell, Hans Christian Beck, Jesper Rømhild Davidsen

**Affiliations:** 1Department of Clinical Immunology, Odense University Hospital, DK-5000 Odense, Denmark; 2South Danish Center for Interstitial Lung Diseases (SCILS) and Pulmonary Aspergillosis Center Denmark (PACD), Department of Respiratory Medicine, Odense University Hospital, DK-5000 Odense, Denmark; christian.b.laursen@rsyd.dk (C.B.L.);; 3Odense Respiratory Research Unit (ODIN), Department of Clinical Research, University of Southern Denmark, DK-5230 Odense, Denmark; 4Department of Clinical Biochemistry and Pharmacology, Centre for Clinical Proteomics, Odense University Hospital, DK-5000 Odense, Denmark; amanda.jessica.campbell@rsyd.dk (A.J.C.); hans.christian.beck@rsyd.dk (H.C.B.)

**Keywords:** chronic pulmonary aspergillosis, bronchoalveolar lavage, proteome analysis, neutrophil degranulation, neutrophil oxidative burst, iron chelation

## Abstract

Neutrophil and (alveolar) macrophage immunity is considered crucial for eliminating *Aspergillus fumigatus*. Data derived from bronchoalveloar lavage (BAL) characterizing the human immuno-pulmonary response to *Aspergillus fumigatus* are non-existent. To obtain a comprehensive picture of the immune pathways involved in chronic pulmonary aspergillosis (CPA), we performed proteome analysis on AL of 9 CPA patients and 17 patients with interstitial lung disease (ILD). The dihydrorhodamine (DHR) test was also performed on BAL and blood neutrophils from CPA patients and compared to blood neutrophils from healthy controls (HCs). BAL from CPA patients primarily contained neutrophils, while ILD BAL was also characterized by a large fraction of lymphocytes; these differences likely reflecting the different immunological etiologies underlying the two disorders. BAL and blood neutrophils from CPA patients displayed the same oxidative burst capacity as HC blood neutrophils. Hence, immune evasion by *Aspergillus* involves other mechanisms than impaired neutrophil oxidative burst capacity per se. CPA BAL was enriched by proteins associated with innate immunity, as well as, more specifically, with neutrophil degranulation, Toll-like receptor 4 signaling, and neutrophil-mediated iron chelation. Our data provide the first comprehensive target organ-derived immune data on the human pulmonary immune response to *Aspergillus fumigatus*.

## 1. Introduction

Chronic pulmonary aspergillosis (CPA) is an overlooked and underdiagnosed disease category caused by inhalation of ubiquitous spores from *Aspergillus* species in patients with structural lung abnormalities or reduced immunity [1]. There are no consistent epidemiological data available on CPA, but it is estimated that there are more than 3,000,000 affected individuals worldwide and that the condition causes 340,000 deaths per year [1]. In a recent Danish register-based study, the incidence rate and prevalence of CPA in Denmark were estimated to be 4.8/100,000 person years and 270/years, respectively [2]. The clinical and radiological presentation of CPA is rarely obvious since CPA most often occurs as a continuum of overlapping syndromes, in which one subtype can transform into another [3,4]. As CPA occurs together with insidious symptoms superimposed on the patient’s already existing pulmonary disease, the CPA diagnosis can easily be missed. This may have fatal consequences, as untreated CPA can lead to invasive disease or pulmonary fibrosis with a high 5-year mortality [1]. Unfortunately, there is no independent marker for detecting CPA and the presence of several concurrent criteria is required to establish a diagnosis. As such, CPA is a challenging “puzzle diagnosis” in which a large number of examinations (radiology, cyto-histopathology, microbiology, biochemistry, and serology) are conducted in order to either indicate or exclude CPA [5]. Hence, the identification of new CPA markers could potentially help develop better tools for diagnosis, monitoring disease trajectory, and treatment response.

The pulmonary response to *Aspergillus* is complex and involves primarily neutrophils and macrophages together with a multitude of other factors [6,7]. The evidence extracted from proteome analysis of *Aspergillus*-infected neutropenic rabbits points to mainly inflammatory and transport proteins [8,9] being differentially “expressed” in bronchoalveolar lavage (BAL) from these animals. Proteome analysis revealed a unique protein profile distinguishing these rabbits from rabbits infected with *Pseudomonas* [8], but also revealed that the BAL protein profile changedwith the therapeutic response [9]. However, since these rabbits were neutropenic they may not be well suited to address the neutrophil response to *Aspergillus* in the lungs. Since the diagnostic and treatment monitoring potential of BAL proteomics has been suggested in *Aspergillus*-infected rabbits [8,9], there is reason to believe that human BAL proteomics could also provide us with potentially useful CPA biomarkers.

## 2. Aim

The aim of this study was to use BAL proteomics to explore the immune pathways involved in CPA and to identify potential new CPA biomarkers.

## 3. Methods

### 3.1. Study Design

In this single-center explorative cohort study, we prospectively included patients with CPA who underwent diagnostic BAL during January 2017 to January 2020.

### 3.2. Setting

In the Region of Southern Denmark, diagnosis and management of CPA and interstitial lung diseases (ILD) is accomplished at the South Danish Center for Interstitial Lung Diseases (SCILS), which is placed at Odense University Hospital (OUH), Denmark. SCILS receives patients suspected of CPA and ILD from the primary sector within the local area of Odense municipality, as well as from all other hospital departments at OUH and within the region, thereby serving as a tertiary specialist center covering an area of 1.23 million inhabitants representing the source population (January 2024). The CPA and ILD investigation program covers chest computed tomography (CT), lung physiological tests, and diagnostic bronchoscopy including BAL for culturing, flow cytometric, and study-specific analyses. BAL was performed according to international standards with the bronchoscope placed into a “wedge position”, meaning that the bronchoscope was advanced into a selected subsegmental bronchus until the segmental lumen was occluded. Hereafter, sterile saline in portions of 50 mL was sequentially installed and recovered until at least 70 mL was obtained for further analyses [10].

### 3.3. Cases and Controls

CPA cases were defined as patients referred to SCILSand were all reviewed at a regional fungus multidisciplinary team discussion meeting (MDD) involving specialists of infectious diseases, microbiology, immunology, hematology, respiratory medicine, thoracic surgery, and radiology. At this MDD, each CPA case was discussed individually according to the diagnostic tests that were performed according to the recommendations in present guidelines on diagnostics and treatment]. The obtained BAL samples were analyzed with an emphasis on neutrophil oxidative burst capacity and immune pathway analysis, and they compared to a control group of patients who had BAL performed as part of examination for unclassifiable ILD within the same study period [5].

### 3.4. DHR-Test for Neutrophil NADPH-Generated Oxidative Burst

In neutrophil granulocytes, the NADPH-generated oxidative conversion of dihydrorhodamine 123 (DHR 123, Sigma-Aldrich, St. Louis, MO, USA, 109244-58-8) into fluorescent rhodamine 123 upon activation with phorbol-12-myristat 13-acetat (PMA, Sigma Aldrich, St. Louis, MO, USA, 16561-29-8) (10 μM) was assessed for both BAL and blood. We constructed two tubes, where BAL or blood was either incubated (15 min in 37 °C water bath) with tube (1) PBS + DHR solution (87 μM) and tube (2) PMA + DHR (87 μM). Samples were subsequently incubated in a 37 °C water bath for a further 15 min. After adding diluted FACS lysing solution (1:10), tubes were kept in darkness for 10 min at room temperature. Samples were centrifuged and the supernatant was pipetted away leaving 50 μL. Lastly, PBS was added to the samples followed by centrifugation. The oxidative conversion of DHR 123 to fluorescent rhodamine 123 was measured utilizing a FACS Canto flow cytometer (BD Biosciences, Franklin Lakes, NJ, USA), identifying neutrophils by forward-side scatter characteristics. The median fluorescence intensities (MFI) in both PBS- and PMA-stimulated tubes for both BAL and blood were compared.

### 3.5. Mass Spectrometry

BAL samples were centrifuged (10,000× *g*/4 °C/20 min) immediately after collection, followed by storage of the supernatant at −80 °C until use. For proteome analysis, the BAL was thawed and centrifuged (10,000× *g*/10 min/4 °C), and the supernatant, -containing proteins, was reduced in the presence of 5 mM of dithiothretiol (DDT) at 50 °C for 30 min, followed by alkylation by iodoacetamide (15 mM, 30 min in the dark). Reduced and alkylated proteins (10 µg per sample) were precipitated by acetone precipitation, re-dissolved in 0.2 M triethyl ammonium bicarbonate, and incubated with 1/50 *w*/*w* trypsin over night at 37 °C. The resulting tryptic peptide samples were labeled using the 16-plex tandem mass tag (TMTpro, ThermoFisher Scientific, Waltham, MA, USA), where a pool of all samples was labeled with the mass tag 126 and served as an internal standard. Tagged peptides were mixed into two mixed peptide samples that were fractionated into 8 fractions using high-pH chromatography previously described [11].

Nano LC-MS/MS analysis of fractionated samples was conducted on an Orbitrap Exploris mass spectrometer (Thermo Fisher Scientific, Bremen, Germany) coupled with a nanoHPLC interface (Dionex UltiMate 3000 nano HPLC, Sunnyvale, CA, USA). The samples (5 μL) were loaded onto a custom-made, fused capillary pre-column (2 cm length, 360 μm OD, 75 μm ID packed with ReproSil Pur C18 3 μm resin (Dr Maish, GmbH, Ammerbuch, Germany)) with a flow of 5 μL/min for 6 min. Trapped peptides were separated on a custom-made, fused capillary column (25 cm length, 360 μm OD, 75 μm ID, packed with ReporSil Pur C13 1.9 μm resin) using a linear gradient ranging from a 91 to 86% solution A (0.1% formic acid) to 25–34% B (80% acetonitrile in 0.1% formic acid) over 100 min, followed by 5 min at 90% B and 5 min at 98% A at a flow rate of 250 nL per minute.

Mass spectra were acquired in positive ion mode, thereby applying an automatic data-dependent switch between an Orbitrap survey MS scan in the mass range of 400–1400 *m*/*z* followed by peptide fragmentation, which was achieved by applying a normalized collisional energy of 38% in a 3 s duty cycle. The target value in the Orbitrap for t MS scan was set to 1 × 10^6^ ions at a resolution of 60,000 at *m*/*z* 200, and 2 × 10^5^ ions at a resolution of 45,000 at *m*/*z* 200 for MS/MS scans. The ion selection threshold was set to 1 × 10^4^ counts. Selected sequenced ions were dynamically excluded for 60 s.

All Exploris raw data files were processed and quantified using Proteome Discoverer version 3.0.1.27 (Thermo Scientific, Waltham, MA, USA). The CHIMERYS (Inferys 2.1, Thermo Fisher Scientific, Waltham, MA, USA) search engine built into Proteome Discoverer was used to search the data using the following criteria—protein database: Uniprot (downloaded 22 February 2023, 20,330 entries) and restricted to humans. Fixed search parameters included trypsin, allowing for two missed cleavages; carbamidomethylation at cysteines; and TMTpro (Thermo) labeling at lysine and N-terminal amines while methionine oxidation was set to dynamic. The mass tolerance of the fragment was adjusted to 20 ppm. Data were also searched with a TMTpro-specific spectral library using MSPepSearch with a precursor and fragment mass tolerance of 15 ppm [12]. A false discovery rate (FDR) was calculated using a decoy database search, and only peptide identifications with high confidence (misrecognition rate < 1%) were included.

### 3.6. Proteome Data Processing and Statistical Analysis

After processing proteomics data with Proteome Discoverer, further analysis was performed in RStudio 2022.07.1 + 554 with R 4.2.1 [13,14]. Proteins with less than 3 observations in either the case or control group were removed, and data were log2-transformed, after which ~85% of the proteins followed a normal distribution, as determined by a Shapiro–Wilk test. In addition, ~90% of the proteins demonstrated equal variance between the case and control groups, as determined by an F-test. The differential protein expression was determined using a two-sided Student’s *t*-test with Benjamini–Hochberg correction for multiple testing and an FDR of 5%. ComplexHeatmap [15] was used to perform hierarchical clustering. Euclidean distance was used to calculate the pairwise dissimilarity while complete linkage was used to measure the dissimilarity between clusters. Human proteins annotated as ‘Response to fungus’ (GO:0009620) were downloaded from QuickGO (GO version 2024-01-28, EMBL-EBI, Wellcome Genome Campus, Hinxton, Cambridgeshire, CB10 1SD, UK) [16]. The STRING database 12.0 was used to investigate enriched pathways [17]. Search settings were set to ‘high confidence’ (0.7) and 5% FDR using all protein–protein interaction sources. Enriched Reactome pathways were exported.

Data on 123 rhodamine formation for each clinical group were given as the median with minimum and maximum values. Comparisonsbetween groups were assessed by a Mann–Whitney U test. The significance level was 5%.

## 4. Results

### 4.1. Patient Population and Baseline Data

Twenty-eight patients, comprising 10 patients with CPA and 18 patients with ILD, were included in the study. The baseline data of the included patients are presented in Table 1.

In the CPA patient group, 7 patients had chronic cavitary pulmonary aspergillosis (CCPA), and 3 had subacute invasive pulmonary aspergillosis (SAIA). In the ILD patient group, 11 patients had idiopatic pulmonary fibrosis (IPF), 3 had non-specific interstitial pneumonia (NSIP), and the remaining 4 patients unclassifiable ILDs. In all included patients, BAL samples were collected as described in the methods section (see Section 3.2).

### 4.2. BAL Leukocyte Composition in Patients with CPA and ILD

Defined by flow cytometric forward-side scatter characteristics, BAL from CPA patients was generally dominated by granulocytes (56.6%: 28.6–79.5%), with generally only a modest presence of lymphocytes (0.9%: 0.0–13.7%). We were not able to delineate monocytes by forward-side scatter characteristics; however, Sysmex analysis of a single CPA BAL showed that monocytes constituted 6.3% of leukocytes followed by lymphocytes at 1.6%, where the remainder were granulocytes and cellular debris. As judged by forward-side scatter characteristics, ILD BAL was constituted by an increase in lymphocytes (32.3%: 4.0–48.6%) compared to granulocytes (33.5%: 16.5–41.2%). The differences in granulocyte (CPA vs. ILD, *p* = 0.03) and lymphocyte (CPA vs. ILD, *p* < 0.001) constitution between the CPA and ILD groups were significant.

### 4.3. DHR-Profiling of BAL-Fluid and Blood Neutrophils from CPA Patients

In the CPA patients, the PMA-stimulated BAL (n = 9, 140,986 MFI; 2249–36,635 MFI) and blood neutrophils (n = 8, 17,519 MFI; 6502–39,861 MFI) did not differ in their 123 rhodamine formation (*p* = 0.74). The PMA-stimulated BAL and blood neutrophils from the CPA patients displayed a similar 123 rhodamine formation compared to the PMA-stimulated blood neutrophils from the healthy controls (n = 8, 13,559 MFI; 5125–22,770 MFI) (*p* = 0.88 for CPA BAL neutrophils versus HC blood neutrophils and *p* = 0.33 for CPA blood neutrophils versus HC blood neutrophils, Figure 1).

### 4.4. Proteome Analysis

A total of 4460 proteins, with at least three observations per experimental group were observed in 26 samples (n = 9 for CPA cases, n = 17 for ILD patients [in 1 CPA and 1 ILD patient, proteome analysis was not performed]), 147 of which were differentially expressed after correction for multiple testing, thereby demonstrating that proteomic differences exist in BAL from patients with CPA compared to BAL from patients with other lung diseases (Appendix A). Most of the proteins were upregulated with only 17 proteins downregulated (Figure 2A). To explore whether protein expression could be used to differentiate between the pulmonary fluid from CPA positive and negative patients, a hierarchical clustering of differentially regulated proteins with no missing values (114/147) revealed that samples could be partially clustered into case and control groups (Figure 2B). One large cluster contained four CPA cases (ID = 2, 7, 9, and 14) and all of the controls. In this cluster, three of the cases along with one control were linked together first before they were joined to the rest of the samples. The second main cluster was composed of the remaining five cases where protein expression was more variable, as demonstrated by the longer branch lengths before the samples were linked.

Next, the differentially expressed proteins were compared to the proteins annotated with the GO term ‘Response to fungus’ (GO:0009620) to identify the key proteins that could potentially be used to achieve a similar separation of the cases and controls. Seven proteins from the data were annotated, namely retinoic acid receptor responder protein 2 (Q99969); myeloperoxidase (P05164); cathepsin G (P08311); protein S100-A8, A9, and A12 (P05109, P06702, and P80511); and cathelicidin antimicrobial peptide (P49913), all of which are known to be involved in the immune system function. The clustering of patients based on the expression levels of these seven proteins alone could also result in a partial separation of cases and controls (Figure 2C). This suggests that immune-related proteomic changes in BAL could be used to distinguish between patients with CPA and patients with other lung diseases.

The reactome pathways confirmed that many of the differentially expressed proteins were involved in immune system function, including neutrophil degranulation and immune activation pathways such as Toll-like receptor cascades (Table 2).

A visual representation of the protein–protein interactions confirmed that the main protein network was largely composed of proteins involved in the innate immune system, specifically those in neutrophil degranulation (Figure 3). For example, proteins S100-A8, A9, and A12 formed antimicrobial protein complexes. Several other antimicrobial proteins were also central to the network, such as azurocidin (AZU1), cathepsin G (CTSG), and bacterial permeability-increasing protein (BPI), among others.

These initial results suggest that the composition of the pulmonary fluid proteome in CPA patients changes in response to *Aspergillus* infection, which could potentially be used to identify these patients.

## 5. Discussion

Innate immunity is considered critical in the defense against pulmonary aspergillosis [18], thus aligning with our observation that innate immunity proteins are enriched in CPA BAL compared to ILD BAL. The pulmonary immune response to *Aspergillus* is complex and involves genetic susceptibility loci [19], MBL-status [20], as well as dynamic components such as prior or present medication [21] and fungal growth kinetics [22]. Additional parameters influencing fungal virulence include underlying pulmonary status (i.e., COPD) [3] and tissue hypoxia [23]. Hence, a systems biology approach, such as proteomics for instance, is well suited to characterize the multiple protein determinants involved in *Aspergillus*-derived immune interactions at a given time point. Neutrophils play a pivotal role in the elimination of *Aspergillus*, as evidenced by the clinical association between severe *Aspergillus* infections and compromised neutrophil immunity such as chronic granulomatous disease (CGD), neutropenia, and a perturbed IL-17 axis [24]. In mice, neutrophils seem to be critical in containing *Aspergillus* at early time points after infection [25], and neutrophils prevent the germination of conidia via complement receptor 3 (CR3) and the focal adhesion protein kindlin 3 while containing hyphae through antibody-induced NADPH oxidase activity [26]. The human neutrophil detection of *Aspergillus* conidia is dependent upon the integrin CD11b/CD18, the latter triggering ROS-independent killing through the PI3K pathway [26]. Evading conidia, having sprouted into hyphae, are opsonized through neutrophil antibody Fcγ receptors, thereby leading to ROS-dependent killing that is mediated through Syk, PI3K, and protein kinase C [26]. The release of neutrophil granular contents and the release of neutrophil extracellular traps also contributes to anti-fungal immunity [27,28,29].

In contrast, lymphocytes play a more prominent role in the pathogenesis of many ILD subtypes, especially in non-fibrotic ILD subtypes that predominate inflammation [30]. Hence, the different immunological etiologies underlying CPA and ILD was likely mirrored in the significantly different leucocyte composition. The BAL from CPA patients was dominated by neutrophils, while lymphocytes constituted a major leukocyte subpopulation in ILD BAL. These differences in BAL leukocyte composition were substantiated by our observation that proteins related to neutrophil degranulation were enriched in CPA BAL (Table 2). The choice of ILD patients as controls was partly dictated by the referral pattern at the SCILS. However, the differential diagnosis to CPA includes Mycobacterium tuberculosis, non-tuberculosis Mycobacteria, histoplasmosis, actinomycosis, coccidioidomycosis, and lung carcinoma [5]. Hence, in order to elucidate the diagnostic potential of BAL proteome analysis with regard to CPA, a more appropriate control group should include patients with pulmonary fungal infections. In mice, one of the non-oxidative ways of eliminating *Aspergillus fumigatus* is through a granular release of neutrophil elastase [31]. Furthermore, human neutrophil specific granules contain lactoferrin, which through its iron-depleting effect, can inhibit the growth of *Aspergillus* [32]. Gazendam et al. also confirmed the central role of non-oxidative intracellular lactoferrin release in the neutrophil killing of *Aspergillus* conida [26]. Interestingly, we found CPA BAL to be enriched by another iron-binding protein lipocalin-2 (LCN2), also called neutrophil gelatinase-associated lipocalin as it is synthesized by neutrophils, which sequesters siderophores [33]. Siderophores are small iron-chelating substances that are used by bacteria and fungi, including *Aspergillus fumigatus*, to sequester essential iron and facilitate growth [34]. By providing direct evidence from human lungs, our proteome data substantiate the previous animal models and in vitro findings, thereby suggesting neutrophil degranulation and iron chelation to also be important in the human pulmonary immune response to *Aspergillus.*

The ability to detect and destroy evading *Aspergillus* conidia, which sprout into hyphae, is dependent on the neutrophil antibody Fcγ receptors leading to ROS-dependent killing mediated through Syk, PI3K, and protein kinase C [26]. We observed that CPA BAL neutrophils generated PMA-induced NADPH-dependent oxidative bursts similar to those observed in neutrophils from the blood of healthy controls. Hence, this is the first study to show that the BAL neutrophils from CPA patients have an unimpeded, NADPH-dependent oxidative response capacity. Thus, immune evasion by *Aspergillus* and the progression to fulminant CPA is likely caused by defects in innate sensing or subsequent innate intracellular downstream signaling to *Aspergillus* conida and hyphae, independent of neutrophil oxidative burst capacity per se.

Our proteome data from BAL support a role for a Toll-like receptor 4 (TLR4)-mediated pulmonary immune response to *Aspergillus*, as proteins associated with the TLR4 pathway were found to be enriched in CPA-BAL (Table 2). To the best of our knowledge, this is the first study directly documenting the involvement of the TLR4-pathway in the human pulmonary immune response directed against *Aspergillus*. TLR4 detection and subsequent signaling through the Myd88 pathway was observed to be important for the elimination of *Aspergillus* in mice [35]. In human embryonic kidney (HEK), 293 cells were transfected with human TLRs, both TLR2 and TLR4 were critical for the detection of *Aspergillus fumigatus* conida and hyphae, as well as for the subsequent recruitment of neutrophils [36]. However, there is also evidence that *Aspergillus fumigatus* tries to subvert detection by TLR2 and TLR4. In in vitro-differentiated human macrophages, *Aspergillus fumigatus* conida and hyphae attenuated TLR4 signaling, and this was not due to decreased expression of TLR4 mRNA [37]. Interestingly, while downmodulating TLR4 responses, *Aspergillus fumigatus* hyphae actually increasedTLR2 responses [37]. Chai et al. hypothesized that the subsequent TLR2/TLR4 imbalance could favor a Th2 response, more permissible for *Aspergillus* survival [37]. Thus, the TLR4 axis seems to be an important fulcrum for *Aspergillus*–host interaction. Cathelicidin antimicrobial peptide (CAMP also termed LL-37) was another protein found to be enriched in CPA BAL compared to ILD BAL. CAMP is found in different leukocyte subsets, including neutrophils and macrophages [38], but can also be present in epithelial cells and keratinocytes. CAMP is generated in these cells in response to different microbes including fungi [39]. CAMP was found to bind directly to the membrane of *Aspergillus* and destroy its integrity [40]. CAMP also reduced *Aspergillus*-induced macrophage inflammation and hence may be protective of host tissue [40]. Machata et al. performed proteome analysis on thawed BAL samples derived from 27 patients, the majority witha probable diagnosis of invasive pulmonary aspergillosis (IPA) and only 15% with a proven diagnosis. To increase the specificity of their protein findings, Machata et al. included a leukopenic mice model of IPA. Their controls consisted of non-IPA patients and healthy mice, respectively [41]. Interestingly, Machata et al. found only a single neutrophil marker, CD177, to be differentially expressed in both men and mice. However, the presence of (severe) neutropenia in some of their patients, as well as the neutropenic status of the IPA mice, might have affected their findings. None of our CPA patients were neutropenic, which highlights the dependency of differential proteomics on patients’ disease characteristics.

It goes without saying that, due to the very small sample size, our data require confirmation on a larger CPA population, preferentially with a control group consisting of patients with fungal or mycobacterial pulmonary infections. We do not expect proteome analysis to supplant the conventional diagnostic modalities, which likely provide superior information on CPA disease severity and extension. Proteome analysis is still more cumbersome and expensive than the conventional diagnostic toolbox for CPA. But proteome analysis on BAL could, due to the detection of CPA-related immune proteins and immune pathways, potentially reveal new immunological insights with respect to disease pathogenesis, prognostic CPA markers, and targets for therapy. The immunological consistency between our human ex vivo data and previous animal and in vitro modeling of anti-*Aspergillus* immunity makes us confident that this exploratory study is a small step toward clarifying these issues.

## 6. Conclusions

Through deriving data directly from the affected target organ, our findings suggest that innate immune-pathways linked to neutrophil activation, neutrophil-mediated iron chelation, and innate immune sensing through the TLR4-pathway are actively involved in CPA and hence may have potential as new diagnostic biomarkers for this serious condition.

## Figures and Tables

**Figure 1 jof-10-00314-f001:**
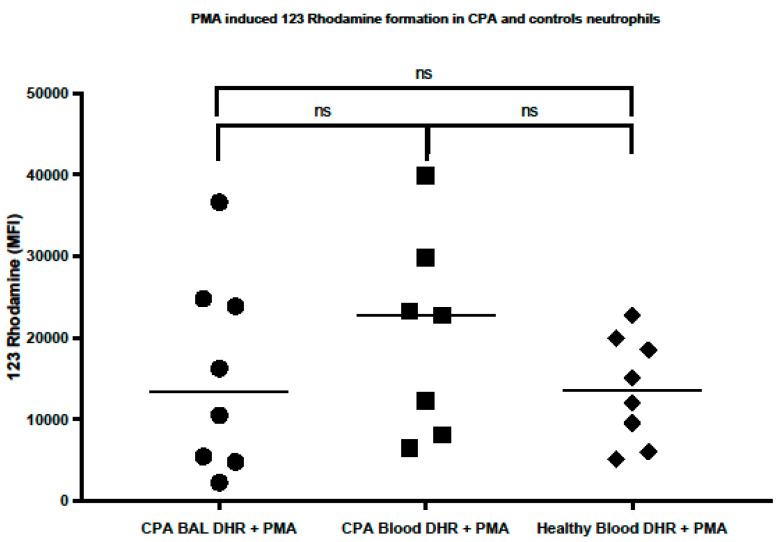
In the PMA-stimulated neutrophils from BAL (CPA patients) and blood (CPA patients or healthy controls), the neutrophil NADPH-generated oxidative burst capacity was measured as conversion of 123 dihydrorhodmaine into 123 rhodamine (where the median fluorescence intensity of 123 rhodamine formation is shown on the *y*-axis). CPA BAL and blood neutrophils showed no signs of impaired neutrophil NADPH-generated oxidative burst capacity.

**Figure 2 jof-10-00314-f002:**
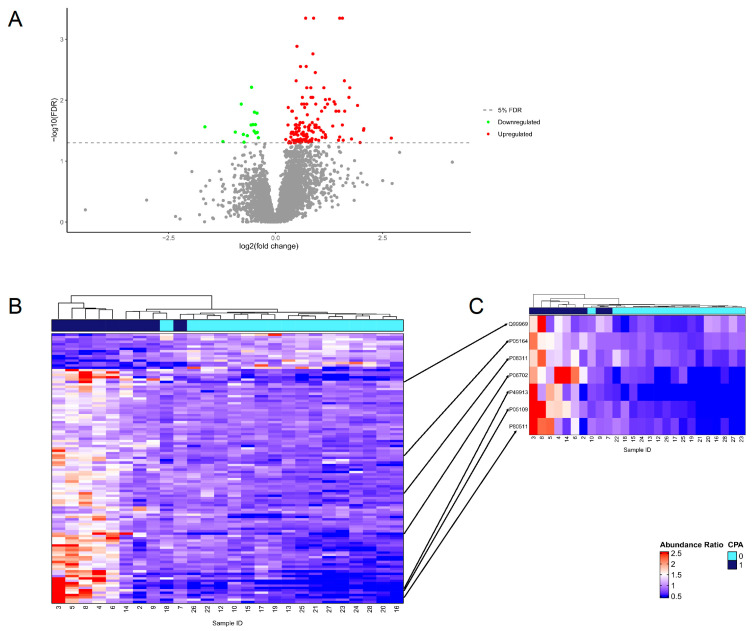
Protein expression in pulmonary fluid. (**A**) Volcano plot of 147 significantly differentially expressed proteins. Upregulated proteins are shown in red and downregulated proteins are shown in green while nonsignificant proteins are shown in gray. The 5% FDR threshold is indicated by a dashed gray line. (**B**) Unsupervised hierarchical clustering and heatmap based off the protein expression of 114 significantly differentially expressed proteins with no missing values. (**C**) Unsupervised hierarchical clustering and heatmap based off seven significantly differentially expressed proteins annotated with ‘response to fungus’ (GO:0009620). The arrows indicate where these proteins can be found in (**B**).

**Figure 3 jof-10-00314-f003:**
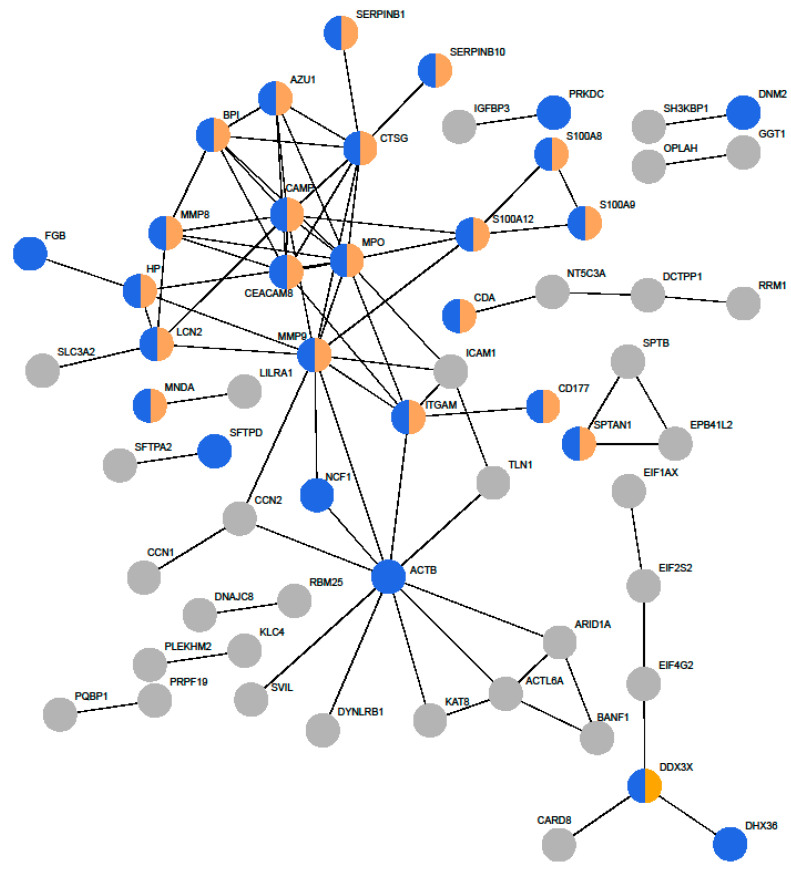
Protein–protein interactions represented by STRING. Differentially expressed proteins were queried with the STRING database. Enriched reactome pathways were used to color code the proteins, with blue designating the proteins involved in the innate immune system (HSA-168249) and blue and yellow designating the proteins specifically involved in neutrophil degranulation (HSA-6798695). For simplicity, proteins are indicated by their gene names. Proteins without any interactors were removed.

**Table 1 jof-10-00314-t001:** Patient characteristics.

	CPA Patients	ILD Patients
Subjects (n)	10	18
Male sex (n (%))	5 (50%)	11 (61%)
Age—years (mean (min/max))	58 (21/83)	72 (55/86)

**Table 2 jof-10-00314-t002:** Selected enriched reactome pathways.

Identifier	Reactome Pathway	Observed Count	Background Count	Strength	FDR	Gene Name
HSA-6798695	Neutrophil degranulation	26	476	0.87	6.73 × 10^−12^	CTSG, MPO, AZU1, MMP8, SERPINB10, CEACAM8, BPI, RHOF, CAMP, FCAR, HP, MNDA, S100A8, S100A12, S100A9, MMP9, LCN2, CDA, NHLRC3, SERPINB1, MGAM, CD177, RAB44, SPTAN1, DDX3X, ITGAM
HSA-168249	Innate immune system	36	1041	0.67	7.07 × 10^−12^	CTSG, MPO, AZU1, MMP8, SERPINB10, CEACAM8, BPI, RHOF, NCF1, CAMP, FGB, PSMC5, PRKDC, UBA7, FCAR, HP, MNDA, S100A8, S100A12, S100A9, SFTPD, MMP9, CNPY3, LCN2, CDA, NHLRC3, SERPINB1, DNM2, DHX36, MGAM, CD177, RAB44, SPTAN1, DDX3X, ACTB, ITGAM
HSA-166016	Toll-like receptor 4 (TLR4) cascade	8	139	0.9	0.0055	BPI, FGB, S100A8, S100A12, S100A9, SFTPD, DNM2, ITGAM
HSA-6799990	Metal sequestration by antimicrobial proteins	3	6	1.84	0.0099	S100A8, S100A9, LCN2
HSA-6803157	Antimicrobial peptides	6	87	0.97	0.017	CTSG, BPI, CAMP, S100A8, S100A9, LCN2

Enriched reactome pathways. Differentially expressed proteins were queried with STRING, and overrepresented reactome pathways were exported. The observed count represents the number of proteins in the query data annotated with the indicated reactome pathway, while the background count indicates all the gene products annotated with the pathway. Strength is the log10 ratio of the observed-to-expected proteins when considering a random protein network of the same size. FDR values were calculated with Benjamini–Hochberg correction.

## Data Availability

The data presented in this study are available on request from the corresponding authors. The data are not publicly available due to Danish legislature. Data access permissions can be obtained for specific purposes.

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
