# Peer review of "Proteome and Dihydrorhodamine Profiling of Bronchoalveolar Lavage in Patients with Chronic Pulmonary Aspergillosis"

_jof, 2024, doi:10.3390/jof10050314_

Round 1

Reviewer 1 Report

This preliminary study derived potential biomarkers for chronic pulmonary aspergillosis (CPA) by comparing detailed proteomes of fresh BAL supernatants from patients with CPA with those from patients with idiopathic pulmonary fibrosis (IPF), based on the rationale that untreated CPA can progress to severe pulmonary fibrosis. The comparison revealed differences in BAL neutrophil/lymphocyte abundance and in expression of proteins associated with innate immunity, in particular, neutrophil degranulation, Toll-like receptor 4 signalling and neutrophil-mediated iron chelation. The work is based on a small sample of 9 patients with CPA and 17 with IPD, collected over 3 years at a Danish Reference Centre for Interstitial Lung Diseases. It is posited that these protein signatures may be useful diagnostic biomarkers for CPA, noting that occurrence of particular proteins correlated with previously reported elements of the complex host response to invasive aspergillosis in both animals and humans. The preliminary observation that the neutrophil oxidative burst is likely normal in BAL supernatants and blood of patients with CPA (compared with blood of 8 healthy controls) is of interest but certainly needs to be substantiated in other studies.

A weakness of the study, as acknowledged by the authors, is the small sample size, and hence the work is presented as a pilot.

Controls: Further explanation of the use of ILD patients as controls for the broad group of patients with CPA, in which it is not clear how many had pulmonary fibrosis would be helpful. Notably, IPF is not typically considered in the differential diagnosis of patients presenting with features of CPA (multiple Denning refs), although in fairness, its incidence/point prevalence in the CPA spectrum is uncertain, especially in resource-poor countries. Furthermore, IPF BALs were dominated by lymphocytes not neutrophils and neutrophil-related proteins formed the basis of the distinguishing features for CPA. It would be desirable to discuss diseases typically considered in the differential diagnosis of CPA including other infections. In the discussion summary, it would be desirable to indicate which diseases/diagnoses should be included as comparator/control groups in future studies.

It would be helpful in a supplementary table to detail the following for each patient: date of presentation, clinical presentation/main symptoms and duration, diagnostic tests used to confirm CPA and which proteins in the potential biomarker groups especially, were significantly changed. This would be especially helpful since SAIA (n=2) and IPA (n=1) patients were included and 4 of the 9 with CPA (not otherwise classified) clustered with controls.

What proportion of patients referred for investigation of CPA/IPF over the three years (who did/did not have BAL) were included in the study? If not 100%, how were patients selected?

The introduction is overly complex, detailed and long. Some content is better suited to the discussion section of the manuscript. Reference 1 estimates CPA occurring post-TB, the epidemiology of which has been updated in a recent (non-cited) publication (2024) viz. Denning DW Global incidence of mortality of severe fungal disease. Lancet Infect Dis 12/1/24 online.

The authors indicate that the current tests on which a diagnosis of CPA is based are numerous i.e. radiology, cyto-histopathology, microbiology, biochemistry and serology, and hence new CPA markers would potentially help develop better tools for diagnosis, disease “monitoration” (I’ve not heard of this word perhaps substitute with “monitoring disease trajectory)” and treatment response, though it is difficult to envisage that a single proteomics based test would replace the need to identify the extent of pulmonary disease, the need for microbiological proof of the presence of aspergillus or an aspergillus- specific immune response (precipitins/IgG antibody) as criteria for diagnosis and for cyto-histopathology to confirm if invasive disease is present. It would certainly be valuable to establish a simple and informative test or combination of tests applicable in both resource-rich and resource-poorer Lower/Middle Income Countries that correlate with disease progression/response to therapy/cure or prognosis. Suggest rewording these sentences to clarify meaning.

It would also be helpful to clarify that rabbit and mouse studies referred to are of invasive aspergillosis in immunosuppressed (neutropenic) animals.

There is one recent reference (not cited) describing proteomic analysis of BAL fluids in humans and mice with invasive aspergillosis (IA, Machata S et al. Virulence (2020) 11(1), 1337-1351) which is worth considering as the human disease (IA) is aligned for the animal model and includes fungal as well as host response proteins. Interestingly this paper used BALs collected and stored at -80 degrees and thawed before processing for proteomic studies. These samples would presumably include contents of cells lysed by freeze thawing of the BAL. They may be a more practical solution in practice, than using fresh BAL supernatants if patients can’t easily be referred to a single reference centre. Hence comparing fresh BAL supernatants with freeze-thawed samples may be of value. Authors’ comment?

Methods and analyses: These are clearly described and sufficiently detailed to be reproduced by others.

Reference 2 is an unreferenced editorial which I suggest should be substituted or omitted.

Reference 7 is the same as reference 31.

The majority of references are more than 5 years “old” and apart from the fact that a number can be deleted by simplifying the introduction, they are generally appropriate.

There are numerous spelling errors throughout the manuscript which I note will be addressed by JoF editorial staff.

P3 line 117-119. Did all patients included have this range of tests?

P3 line127. “ERS” should be spelled out before using abbreviation.

P6 lines 251-252. Which diagnostic CPA category did the 4 CPA patients clustered with the controls belong to?

P10 lines 325-327. Please clarify the meaning of this sentence

Author Response

Dear Reviewer 1.

Thank You for Your critical and constructive comments which we will try to address in the following

1) We have now shortened the Introduction considerably, removing most of the material, also now shortened, from the Introduction to the Discussion (line 310-330). We hope this has improved the readability of the Introduction  and Discussion.

 2) We fully agree with the critical problem of sample size. However this study was intended solely as explorative, as especially the recruitment of sufficient CPA patients was challenging. However, the involvement of different parts of neutrophil (and TLR4) immunity, for the first time detected in human CPA lungs, aligns with previous in-vitro data and animal models on the role of the neutrophil in Aspergillus detection. Hence, we think that our data suggest that neutrophil activation markers may have diagnostic and prognostic potential in (non-neutropenic) human CPA patients. We have now stressed the small sample size and the explorative purpose in the Discussion, line 400 and 409 (all line references to the manuscript version with marked changes)

3) Reviewer comment: Controls: Further explanation of the use of ILD patients as controls for the broad group of patients with CPA, in which it is not clear how many had pulmonary fibrosis would be helpful. Notably, IPF is not typically considered in the differential diagnosis of patients presenting with features of CPA (multiple Denning refs), although in fairness, its incidence/point prevalence in the CPA spectrum is uncertain, especially in resource-poor countries. Furthermore, IPF BALs were dominated by lymphocytes not neutrophils and neutrophil-related proteins formed the basis of the distinguishing features for CPA. It would be desirable to discuss diseases typically considered in the differential diagnosis of CPA including other infections. In the discussion summary, it would be desirable to indicate which diseases/diagnoses should be included as comparator/control groups in future studies.

Reply:

However, the main reason for choosing ILD patients as controls were due to the logistic set-up possible, where we chose to use rare disease categories that would undergo almost identical invasive examination regimes as performed in South Danish Center for Interstitial Lung Diseases (SCILS) at Odense University Hospital (OUH), Denmark. Exactly this applies to CPA and ILD. 

We have now added the following to the Discussion (line 338-343):"The choice of ILD patients as controls was partly dictated by the referral pattern at SCILS. However, the differential diagnosis to CPA includes Mycobacterium tuberculosis, non-tuberculosis Mycobacteria, histoplasmosis, actinomycosis, coccidioidomycosis and lung carcinoma (31). Hence, in order to elucidate the diagnostic potential of BAL proteome analysis with regard to CPA, a more appropriate control group should include patients with pulmonary fungal infections. "  

4) Reviewer comment: It would be helpful in a supplementary table to detail the following for each patient: date of presentation, clinical presentation/main symptoms and duration, diagnostic tests used to confirm CPA and which proteins in the potential biomarker groups especially, were significantly changed. This would be especially helpful since SAIA (n=2) and IPA (n=1) patients were included and 4 of the 9 with CPA (not otherwise classified) clustered with controls.

Reply:

We thank the reviewer for this point. All CPA patients were reviewed at a prior regional fungus multidisciplinary team discussion meeting (MDD) involving specialist within infectious diseases, microbiology, immunology, hematology, respiratory medicine, thoracic surgery, and radiology. At this MDD, each CPA case was discussed individually according to diagnostic tests performed according to recommendations in present guidelines on diagnostics and treatment (ref.: Denning D, ERJ 2016 (PMID: 26699723)). Consequently, we believe that the actual CPA diagnoses are well grounded, and that the available information in Table 1 therefore is valid for the purpose of this explorative study.

We fully understand the point on including IPA in a CPA group which regards to a MDD consensus decision, where a “mixed” diagnosis of both SAIA and probable IPA was considered. In such, the MDD conclusion was “IPA/SAIA.” Retrospectively and after having re-reviewed the case among authors, we have agreed on that the patient was misclassified as having IPA, and that the most appropriate diagnosis would have been SAIA. In such, we have changed the wording in the Results section, page 5, line 216:"and three had subacute invasive pulmonary aspergillosis (SAIA)"

5) Reviewer comment: What proportion of patients referred for investigation of CPA/IPF over the three years (who did/did not have BAL) were included in the study? If not 100%, how were patients selected?

Reply:

We presume this question regards to BAL as part of proteome investigation in this study.

This is a relevant question, which we unfortunately cannot answer in detail, as we do not have available data on the precise number of patients from the two disease categories. This applies to changes in our electronic patient journals, and that both patient categories unfortunately have not been consistently and sufficiently ICD-10 coded. In such, we not have the opportunity to estimate the denominator corresponding to the number of patients for CPA and ILD, which is a prerequisite for giving an overall accurate estimate of the proportion of patients referred for investigation.

During the study period, newly diagnosed CPA patients were asked to be part of the study, but many declined participation in a study involving an invasive diagnostic procedure, and further more CPA patients where not candidates at all due to severe lung disease and severe comorbidity.

6) We have qualified the section in the Introduction dealing with the two proteome studies in rodents (line 101-102): " However, since the rabbits were neutropenic they may not be well suited to address the neutrophil response to Aspergillus in the lungs"

7) In the Introduction (line 52-53), disease monitoration has been replaced by:" monitoring disease trajectory" as suggested.

8) At the end of the Discussion, we have now tried to contextualize the role of BAL proteomics in CPA diagnosis/prognosis (line 402-408):" We do not expect proteome analysis to supplant the conventional diagnostic modalities which likely provide superior information on CPA disease severity and extension. Proteome analysis is still more cumbersome and expensive than the conventional diagnostic toolbox for CPA. But proteome analysis on BAL could, due to the detection of CPA related immune proteins and immune pathways, potentially reveal new immunological insights as regards disease pathogeneisis, prognostic CPA markers and targets for therapy."

9) We understand why reviewer 1 had the impression we used fresh BAL samples. However, we used frozen BAL samples,- like Machata. That has now been explicitly stated in ther methods section (line 152-153): "BAL samples were centrifuged (10,000 x g/ 4  ÌŠC/20 min) immediately after collection followed by storage of the supernatant at -80  ÌŠC until use."

10) We have now included the study by Machata et al in the Discussion (line 389-398):"Machata et al performed proteome analysis on thawed BAL samples derived from 27 patients, the majority with a probable diagnosis of invasive pulmonary aspergillosis (IPA) and only 15% with a proven. To increase specificity of their protein findings a leukopenic mice model of IPA was included. Controls consisted of non- IPA patients and healthy mice respectively (51). Interestingly, Machata et al found among proteins differentially expressed in both men and mice only a single neutrophil marker CD177. However, the presence of (severe) neutropenia in some of their patients as well as the neutropenic status of their IPA mice might have affected their findings. None of our CPA patients were neutropenic, which highlights the dependency of differential proteomics on patients’ disease characteristics"

"

11) Reference 2 (Editorial) has now been omitted as suggested.

12) The duplication of the former reference 7 has now been omitted. 

13) We have in this review process discarded most of the references not directly dealing with the roles of neutrophils in Aspergillus immunity

14) We have now included the reference: "Denning DW Global incidence of mortality of severe fungal disease. Lancet Infect Dis"

15) We have deleted 3 of the former references: {Stevens, 2009 #123}, {Romani, 1997 #19} and {Erwig, 2016 #11} and replaced them with {Lionakis, 2023 #204} and {Delliere, 2023 #205} in line 96 (version with marked changes).

Reviewer 2 Report

In the manuscript, the authors elucidated that the potential of diagnostic and therapeutic monitoring by rabbit-derived alveolar lavage proteomics analysis has been demonstrated in Aspergillus infections, and hypothesized that the markers in bronchoalveolar lavage (BAL) could also be utilized.      

The immune pathway of chronic pulmonary aspergillus disease (CPA) was analyzed by proteomics to identify potential new biomarkers of CPA and provide a theoretical basis and new ideas for the diagnosis and treatment of CPA.

This study analyzed the proteome of BAL from patients with CPA to identify the prominently expressed proteins. It is novel that the proteins could serve as potential targets for rapid diagnosis and treatment of CPA. Additionally, a dihydrorhodamine (DHR) assay was performed on BAL and blood neutrophils from CPA patients and compared with those from healthy controls. This research provides novel insights into the immune response of the human lung to Aspergillus fumigatus. It is recommended that the authors should include a discussion on the advantages and disadvantages of using proteomics analysis for CPA diagnosis compared with other methods. It could involve information such as specimen type, sensitivity, specificity, and detection time. By doing so, the study can provide a more comprehensive understanding on how proteomics analysis compares to existing diagnostic approaches for CPA. In addition, the references are relatively old, so it is suggested that the authors should understand the progress of relevant research in recent years to better highlight the advantages of this study.

1. The authors need to provide the informed consents and ethical certification materials of patients with BAL.

2. The authors mentioned the proteome analysis was performed with BAL from 9 CPA patients and 19 ILD patients in the abstract. But in Table 1, there were 10 CPA patients and 18 ILD patients. The authors should verify the data information.

It is recommended that the authors should include a discussion on the advantages and disadvantages of using proteomics analysis for CPA diagnosis compared with other methods. It could involve information such as specimen type, sensitivity, specificity, and detection time. By doing so, the study can provide a more comprehensive understanding on how proteomics analysis compares to existing diagnostic approaches for CPA. In addition, the references are relatively old, so it is suggested that the authors should understand the progress of relevant research in recent years to better highlight the advantages of this study

Author Response

Dear Reviewer 

We are gratefull for Your constructive comments which we will try to address in the following:

1) Reviewers comments: The authors need to provide the informed consents and ethical certification materials of patients with BAL.

Reply:

Please refer to section “Institutional Review Board Statement” and “Informed Consent Statement” at page 12, where these details are already mentioned. However, we have supplemented with following changes in the “Informed Consent Statement” section (lin 426-427):“Written informed consent to participate in this study and to publish the results has been obtained from all patients.

2) Reviewers comments: The authors mentioned the proteome analysis was performed with BAL from 9 CPA patients and 19 ILD patients in the abstract. But in Table 1, there were 10 CPA patients and 18 ILD patients. The authors should verify the data information.

Reply: We have corrected the numbers of CPA and ILD patients in the Abstract, now consistent with the numbers given in the Results (lin 249) : 9 CPA patients and 17 ILD patients. The reason for the incongruence with the numbers listed in Table 1 was due to the fact, that in 1 CPA and 1 ILD patient, no proteom analysis was performed, while the neutrophil oxidative burst analysis was. Hence, only in 9 CPA and 17 ILD patients proteome analysis was performed.

We have added a note (line 249-250):"(......in 1 CPA and in 1 ILD patient, proteome analysis was not performed while neutrophil oxidative burst analysis was, hence 9 CPA cases and 17 ILD patients)" 

3) Reviewers comment: It is recommended that the authors should include a discussion on the advantages and disadvantages of using proteomics analysis for CPA diagnosis compared with other methods. It could involve information such as specimen type, sensitivity, specificity, and detection time. By doing so, the study can provide a more comprehensive understanding on how proteomics analysis compares to existing diagnostic approaches for CPA. In addition, the references are relatively old, so it is suggested that the authors should understand the progress of relevant research in recent years to better highlight the advantages of this study

Reply: In the Discussion we have tried to address reviewer 2's concerns related to the advantages and disadvantages of proteome analysis in comparison to other CPA related diagnostic modalities (line 401-407):"We do not expect proteome analysis to supplant the conventional diagnostic modalities which likely provide superior information on CPA disease severity and extension. Proteome analysis is still more cumbersome and expensive than the conventional diagnostic toolbox for CPA. But proteome analysis on BAL could, due to the detection of CPA related immune proteins and immune pathways, potentially reveal new immunological insights as regards disease pathogeneisis, prognostic CPA markers and targets for therapy."

We have deleted 3 of the former references: {Stevens, 2009 #123}, {Romani, 1997 #19} and {Erwig, 2016 #11} and replaced them with {Lionakis, 2023 #204} and {Delliere, 2023 #205} in line 96 (version with marked changes).

Round 2

Reviewer 1 Report

This assessment is based on a revised version of an original paper.

The authors have made major improvements and have largely addressed all my questions to my satisfaction. They have provided appropriate caveats to their work as requested.

The authors have justified not including a detailed (supplementary) table of individual cases by providing extra comment on the criteria met for CPA case diagnosis and the consensus reached by a multidisciplinary group of experts.

The reason I had suggested inclusion of more detailed information by case relates to the fact that 4 of the CPA cases clustered with controls and five did not ("showing more variation"). Despite the small numbers I wondered whether all four met the criteria for one or other subset of CPA or the group was "mixed" and whether other factors such as time from diagnosis (and hence, presumably, later stage disease) or aspergillus treatment or other factors might be flagged by such a comparison.  This is a relatively minor issue over all but I believe is worthy at least of  comment, if inclusion in a table is not possible.

The authors have inferred that the small number of CPA cases is likely because only those who underwent BAL over the years of study were included. Although in theory this could lead to selection bias, I accept that, in relation to this point, the current manuscript presents new useful and interesting information that is worthy of publication.

It is also interesting that CD177 was picked up in both the Machata study and this one.

Author Response

Dear Editor

We are grateful for having the opportunity to respond to the remaining minor issues raised by reviewer1:

1) In the uploaded new version (language corrected), we have made a few linguistic corrections carried out by our co-author Amanda, who is English.

2) We have hopefully addressed the question concerning more detailed information by case with the following addition (line 92-96):"and were all reviewed at a regional fungus multidisciplinary team discussion meeting (MDD) involving specialist within infectious diseases, microbiology, immunology, hematology, respiratory medicine, thoracic surgery, and radiology. At this MDD, each CPA case was discussed individually according to diagnostic tests performed according to recommendations in present guidelines on diagnostics and treatment "

With best regards

Kristian Assing, corresponding author